# Rational design of aptamer switches with programmable pH response

Ian A. P. Thompson [1], Liwei Zheng[1], Michael Eisenstein[1,2,3] & H. Tom Soh [1,2,3]✉

Aptamer switches that respond sensitively to pH could enhance control over molecular devices, improving their diagnostic and therapeutic efficacy. Previous designs have inserted pH-sensitive DNA motifs into aptamer sequences. Unfortunately, their performance was limited by the motifs' intrinsic pH-responses and could not be tuned to operate across arbitrary pH ranges. Here, we present a methodology for converting virtually any aptamer into a molecular switch with pH-selective binding properties — in acidic, neutral, or alkaline conditions. Our design inserts two orthogonal motifs that can be manipulated in parallel to tune pH-sensitivity without altering the aptamer sequence itself. From a single ATP aptamer, we engineer pH-controlled target binding under diverse conditions, achieving pH-induced selectivity in affinity of up to 1,000-fold. Importantly, we demonstrate the design of tightly regulated aptamers with strong target affinity over only a narrow pH range. Our approach offers a highly generalizable strategy for integrating pH-responsiveness into molecular devices.

[1] Department of Electrical Engineering, Stanford University, Stanford, CA 94305, USA. [2] Department of Radiology, Stanford University, Stanford, CA 94305, USA. [3] Chan Zuckerberg Biohub, San Francisco, CA 94158, USA. ✉email: tsoh@stanford.edu

Physiological pH conditions are maintained through tight homeostatic control within cells and tissues, and local variations in pH play a critical role in a number of important biological processes[1,2]. For example, in cancer, metabolic changes lead to dysregulation of both intracellular and extracellular pH, creating conditions favorable to cancer cell survival, proliferation, and metastasis[3]. Accordingly, there is considerable interest in engineering molecular systems that exhibit programmed behavior in response to small changes in pH. Such mechanisms could enable precise control of molecular devices for a wide range of clinical applications. For example, one can couple a cell-specific targeting moiety to a pH-sensitive cargo transport domain, where the destination tissue exhibits local variations in pH conditions that selectively induce the release of a molecular payload for imaging or therapy[4–6]. In particular, a number of studies have demonstrated the potential utility of DNA aptamer-based molecular "switches" that undergo a large conformational rearrangement in response to ligand binding or changes in environmental pH[7–9], enabling quantitative in situ imaging[10,11], programmed tumor targeting[12], and rapid clinical diagnostics[13].

However, it remains challenging to generate high-specificity aptamer switches that selectively undergo a strong change in target affinity under specific pH conditions. Most of such efforts to date have entailed the rational design of molecules that couple an existing aptamer to a known pH-responsive domain based on prior knowledge of the aptamer structure. For example, the Ricci group modified a cocaine-binding aptamer with an intramolecular DNA triplex that directly competes with hairpin formation in the native aptamer structure. The resulting aptamer bound cocaine with 100-fold weaker affinity at pH 4 than at pH 7[14]. The DeRosa group used noncanonical base pairing between guanine and protonated adenine to disrupt G-quadruplex formation in a thrombin aptamer, enabling reversible pH-mediated binding of thrombin at pH 7 followed by release at pH 5[15]. Most recently, the Tan group employed a cytosine-rich i-motif structure to preferentially stabilize folding of a tyrosine kinase-7 aptamer in acidic conditions, demonstrating selective binding to the aptamer's membrane-bound target at low pH[16]. However, all these approaches entail the use of defined DNA motifs that alter the aptamer binding structure under very particular pH conditions. The resulting aptamer switches cannot readily be modified to work outside this intrinsic pH-response range or be tuned to adjust the sensitivity of their response, greatly limiting their utility. Furthermore, these approaches are not necessarily generalizable across aptamers, and require a new design process for each novel construct.

We describe here a general strategy that should make it possible to convert virtually any aptamer into a pH-responsive switch that can be tuned to selectively undergo a dramatic shift in affinity under acidic, neutral, or alkaline conditions. We achieve this pH specificity by introducing two orthogonal modes of control that can be manipulated in parallel to tune sensitivity to different pH conditions without altering the core sequence of the aptamer itself. Our approach builds on the intramolecular strand displacement aptamer design, in which an aptamer is covalently linked to a partially complementary displacement strand (DS) through an inert DNA linker[7]. We indirectly manipulate aptamer pH dependence by inserting pH-responsive DNA motifs into either the linker or the DS. In comparison to previously described motif-based approaches, this design minimizes sequence constraints and enables broader tuning of our design. Using this strategy, we have generated an array of pH-responsive strand displacement (PSD) aptamer switches, all based on the same core aptamer sequence, that have been engineered to preferentially bind or release their molecular target under various pH conditions. Our most responsive designs exhibit up to a 1000-fold change in affinity over the narrow range between pH 5.5 and pH 7, greatly exceeding the sensitivity of previously reported pH-responsive aptamer switch designs. In addition to generating switches that are broadly responsive to low or high pH, we also describe the design of a PSD switch, the affinity of which, unlike existing pH-responsive aptamers, undergoes a significant enhancement within a narrow pH "window." To our knowledge, this represents the first description of an engineering strategy for producing such constructs. Since the aptamer sequence itself is not being manipulated in these constructs, our design strategy is broadly applicable to a wide range of aptamers, allowing for the ready integration of pH response into a variety of aptamer-based molecular devices.

## Results

**A linker-based design for binding at low pH.** Our PSD constructs couple an existing aptamer to a complementary DS domain via an internal linker domain. By manipulating the sequences of these two domains, we can introduce two non-interfering methods for controlling the response to different pH conditions without altering the aptamer sequence itself. Initially, we focused on manipulations that confer selective binding under acidic conditions. We theorized that by inserting an intramolecular triplex motif[17,18] into the linker domain (Fig. 1a), we could cause the aptamer to selectively bind only at low pH. At neutral pH, we predicted that the mostly unprotonated linker would adopt a duplexed structure through Watson–Crick base pairing, positioning the aptamer and DS in proximity to each other at either end of the stem-loop (Fig. 1b, top). The closely confined DS (high $[DS]_{eff}$) would then compete strongly with target binding, leading to a low effective affinity for the aptamer switch (high $K_D^{eff}$). At lower pH, increasing protonation of cytosine bases favors Hoogsteen base pairing within CGC triplets in the linker, stabilizing intramolecular triplex formation (Fig. 1b, bottom). In this state, the DS and aptamer should be positioned at either end of the rigid triplex, greatly inhibiting their interaction (low $[DS]_{eff}$). This would decrease binding competition, leading to higher effective affinity of the aptamer switch (low $K_D^{eff}$). Such a construct should therefore prove suitable for selectively binding ligands in acidic conditions, such as solid tumor microenvironments or within the endosomal pathway, allowing targeted delivery of drugs or imaging contrast agents.

To test this design strategy, we used an ATP aptamer[19] as a model system. We designed a variety of PSD constructs in which the ATP aptamer was coupled to a seven-nucleotide (nt) DS by a linker domain containing different intramolecular triplex motif designs. A fluorophore and quencher, positioned at the 3' and 5' ends of the construct, respectively, enabled fluorescence-based quantification of DS hybridization as a proxy for target binding. In our initial construct, TAT60, the linker consisted of a triplex containing six TAT triplets and four CGC triplets as well as internal loops of four and five nt, with five-nt spacers joining the linker to both the aptamer and DS (Fig. 1a). We minimized undesired interactions between the native aptamer and the inserted motif by optimizing the unconstrained loop and spacer sequences (see "Methods" for design details)[20].

To evaluate the pH dependence of target binding for TAT60, we measured the fluorescence change associated with increasing concentrations of ATP over a physiological and near-physiological pH range (4.5–8.5). As predicted, we observed strongly pH-dependent shifts in the ATP binding curve associated with insertion of the triplex motif, with $K_D^{eff}$ decreasing more than 1000-fold in moving from pH 8.5 ($K_D^{eff} = 10$ mM, 95% CI [6.7–19 mM]) to pH 5 ($K_D^{eff} = 2.6$ μM, 95% CI [0.53–11 μM]). Increases in affinity are coupled to increased background signal at low pH, matching the predicted effects of inhibited DS

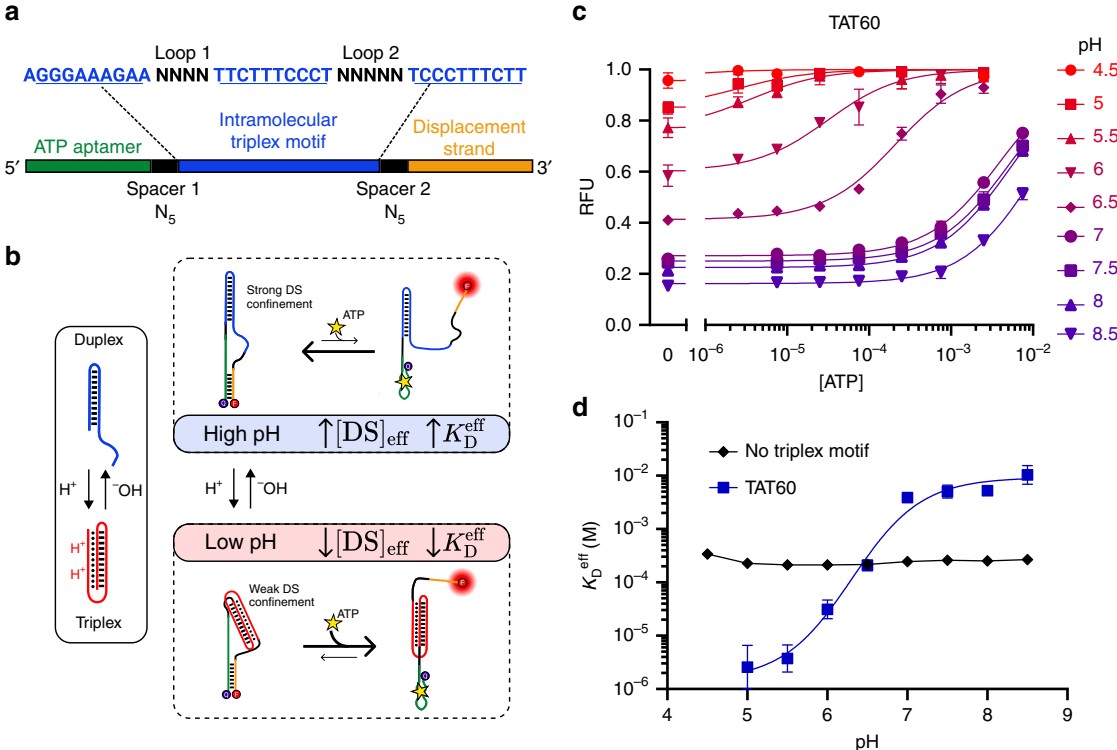

**Fig. 1 Design of an ATP aptamer for selective binding at low pH. a** An intramolecular triplex motif is inserted into the linker domain, which connects the ATP aptamer and displacement strand (DS). The unconstrained loop and spacer sequences are optimized to minimize interference with the aptamer secondary structure. The construct shown here is TAT60, reflecting the 60% TAT composition of the triplex motif. **b** A simplified model for pH-dependent binding. pH dependence is achieved via conformational changes in the triplex motif, which increase the effective DS concentration near the aptamer ($[DS]_{eff}$) at high pH, thereby inhibiting ATP binding and fluorescent signaling. A complete three-state model of the pH-dependent binding mechanism is given in Supplementary Fig. 3. **c** ATP binding curves for the TAT60 construct show a strongly pH-dependent shift in affinity. **d** A standard ATP intramolecular strand displacement aptamer construct with no triplex motif exhibits no pH sensitivity, whereas insertion of the triplex-based linker induces dramatic pH sensitivity. Data points and error bars in **c** show the means and standard deviations of $n = 3$ independent experiments. Data points and error bars in **d** show the best fit values and standard deviations extracted from Langmuir isotherm fits to binding curve data from $n = 3$ independent experiments using nonlinear least squares fitting. Source data are provided as a Source Data file.

hybridization (Fig. 1c, Supplementary Fig. 1). The construct showed especially high sensitivity to pH changes in the physiologically relevant range, as lowering the pH from 7 to 6 produced a 100-fold increase in affinity, with $K_D^{eff} = 3.9$ mM at pH 7 (95% CI [3.2–4.8 mM]) and $K_D^{eff} = 31$ μM at pH 6 (95% CI [19–53 μM]) (Fig. 1d). In contrast, a construct lacking the triplex motif showed no meaningful pH sensitivity, highlighting the effectiveness of our design.

We were able to widely vary the composition of TAT or CGC triplets in the linker without impacting the sequence or secondary structure of the aptamer itself, and this, in turn, allowed us to fine-tune the strength and effective range of the construct's pH dependence. Previously, the Ricci group showed that they could tune the pH range of the duplex-to-triplex transition by altering intramolecular triplex sequence composition, with increased TAT content shifting the transition to higher pH[21]. We exploited a similar strategy, designing a range of constructs with triplex TAT content varying from 50 to 80% to modulate both the dynamic range of pH sensitivity and the magnitude of the pH-dependent affinity change (Fig. 2a). We observed strong pH dependence in all constructs and noted that increasing TAT composition shifted the effective range of the pH-driven transition from acidic to more neutral pH. We found that the transition midpoint, $pK_a$, could be shifted by almost 1 full pH unit by increasing the triplex TAT content from 50% ($pK_a = 6.1$, 95% CI [5.9–6.2]) to 80% ($pK_a = 6.9$, 95% CI [6.5–7.4]) (Fig. 2b). To confirm this

observation, we measured the $pK_a$ of triplex folding in the absence of ATP and observed similar trends of sequence-dependent tuning (Supplementary Fig. 2). Notably, the shift in $pK_a$ induced by a single TAT/CGC substitution within the triplex motif appears to be smaller for our constructs than previously reported values from standalone intramolecular triplex motifs where similar sequence alterations shifted the transition pH from ~6.5 to ~8.5[21]. This smaller per-substitution shift in $pK_a$ should offer finer-grained control over the construct's pH response.

It should also be noted that in addition to decreasing the pH transition midpoint, replacing TAT triplets with protonation-dependent CGC triplets increases the pH dependence of triplex stability, thereby increasing the magnitude of the pH response[22,23]. For example, a triplex motif with 20% CGC content produces a ~42-fold difference in binding affinity between pH 5 and 8 (Fig. 2a, TAT80). By increasing the CGC content to 30% or 40%, we were able to dramatically increase the pH-dependent affinity change over the same range to ~600- and ~2000-fold, respectively (Fig. 2a, TAT70 and TAT60). This tunability allows for the design of aptamer switches that can respond in a finely controlled manner to a broad range of pH conditions.

**A DS-based design for binding at high pH.** We hypothesized that it should be feasible to apply a similar engineering strategy to develop DNA switches that preferentially exhibit high target affinity at neutral or alkaline pH. Such switches could be designed

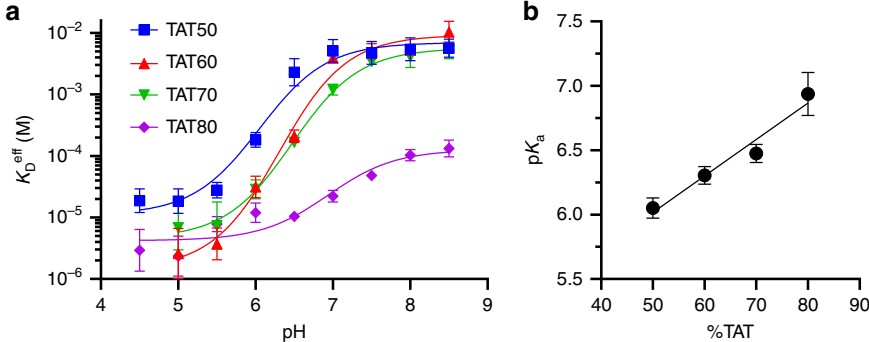

**Fig. 2 pH response tuning through triplex base composition. a** Constructs with varying TAT triplet compositions produce a range of pH dependencies, achieving selective binding at low pH but with varying magnitude of response and pH transition midpoint ($pK_a$). **b** Increasing the TAT content of the triplex motif predictably shifts the $pK_a$ of the switch toward more neutral pH. Data points and error bars in **a** and **b** show the best fit values and standard deviations extracted from fits to binding curve data from $n = 3$ independent experiments (data given in Supplementary Fig. 4) using nonlinear least squares fitting. Source data are provided as a Source Data file.

to bind strongly to a specific ligand at neutral pH, and to release this molecular cargo only upon entering the relatively acidic conditions found in endosomes or tumor microenvironments. In contrast to the linker-based modification we used to induce low pH binding, we established a distinct and orthogonal control mechanism in which we engineered the DS sequence to achieve selective binding at high pH (Fig. 3a). Our group has previously demonstrated that by inserting mismatches within the DS of an intramolecular strand-displacement-based molecular switch, we could decrease DS competitive binding strength ($K_D^{DS}$) and thereby increase the construct's affinity by over 100-fold[24]. While most mismatched bases have negligible interaction strength, certain pH-responsive DNA motifs exploit either A–C or A–G mismatches, which exhibit pH-dependent changes in non-canonical base-pairing strength. Here, protonation of adenine at low pH enables the formation of a second hydrogen bond with either cytosine or guanine, promoting hybridization between these mismatched bases through either an A(H$^+$)–C wobble or an A(H$^+$)–G Hoogsteen base pair[25,26]. We hypothesized that strategic insertion of either A–G or A–C mismatches within the DS would lead to stronger hybridization (increased $K_D^{DS}$) at low pH compared with high pH, thereby selectively increasing the aptamer's ATP affinity at higher pH (Fig. 3b).

We designed two PSD constructs with a non-pH-responsive poly-T linker and varied their DS sequences to include either an A–C or an A–G mismatch. To maximize the pH-response effect, we inserted the mismatch near the middle of the DS, placing the A–C or A–G mismatch four or six nt from the 3′ end, respectively (Fig. 3a). Both constructs showed pH-dependent ATP binding with higher affinity under more alkaline pH conditions, matching our predictions for the effect of increasing the DS hybridization strength (Fig. 3c, Supplementary Fig. 1). A single pH-dependent A–C mismatch within the DS yielded a ~19-fold change in affinity, with the $K_D^{eff}$ increasing from 0.30 mM at pH 8 (95% CI [0.25–0.36 mM]) to 5.6 mM at pH 5 (95% CI [3.7–9.9 mM]) (Fig. 3d) and a transition midpoint of $pK_a = 5.9$ (95% CI [5.8–6.1]). A single A–G mismatch achieved similar pH sensitivity, with an approximately tenfold difference in affinity between $K_D^{eff} = 0.32$ mM at pH 7.5 (95% CI [0.27–0.37 mM]) and $K_D^{eff} = 3.2$ mM at pH 4.5 (95% CI [2.4–4.7 mM]) with $pK_a < 5$. The more acidic transition midpoint pH of these constructs makes them potentially suitable for targeting the endosome-lysosome pathway[27]. The freedom to utilize either A–C or A–G mismatches to induce pH dependence greatly increases the generality of this method. Because these mismatches can be inserted at any base within the DS that base-pairs with an A, C, or

G base within the parent aptamer, our approach is compatible with almost any aptamer sequence.

**Design for selective binding within a narrow pH window.** Since the two tuning strategies described above are non-interfering—with one achieved through linker modification and the other achieved through DS modification—we reasoned that they could be applied in tandem to achieve even greater control over the aptamer pH response. To investigate this capability, we designed a construct that we predicted would exhibit high affinity within only a narrow window of pH range, with greatly reduced binding at both higher and lower pH. Our design integrates both an intramolecular triplex sequence within the linker domain (TAT80) and an A–C mismatch four nt from the 3′ end of the construct (Fig. 4a). These two modifications contribute opposite pH dependencies, but by carefully selecting their pH transition midpoints, we can introduce three-state pH dependence (Fig. 4b). We selected a linker TAT composition that yields a transition midpoint pH higher than that of the A–C-mismatched DS. At high pH, above the transition midpoint of both pH-dependent modifications, the linker forms a duplex that confines the mismatched DS near the aptamer, which leads to low target affinity. Within the narrow range below the linker transition pH but above the DS transition pH, the combined effect of triplex formation and weak DS hybridization decrease binding competition, enabling aptamer folding and high affinity target binding. Finally, as the pH is further decreased below the point of DS transition and the A–C mismatch becomes protonated, DS hybridization is strengthened, inhibiting aptamer folding and favoring release of bound molecular cargo. We combined the TAT80 linker and the A–C mismatch because their individual effects showed a similar magnitude of pH-dependent affinity change over the relevant pH range of 5–8, with a ~19-fold change for the DS modification and a ~42-fold change for the linker modification.

As anticipated, the resulting construct achieves near-native aptamer binding affinity only at the physiological pH range of 6–7 ($K_D^{eff} = 32$ μM at pH 6.5, 95% CI [27–39 μM]) with reduced affinity both at higher pH ($K_D^{eff} = 410$ μM at pH 8.5, 95% CI [330–510 μM]) or lower pH ($K_D^{eff} = 110$ μM at pH 5, 95% CI [87–140 μM]) (Fig. 4c). Importantly, the individual pH dependencies of the linker and DS modifications closely predict the overall pH dependence of the combined construct at either pH extreme, indicating that it should generally be straightforward to design such constructs based on the measured behavior of either modification alone (Fig. 4d).

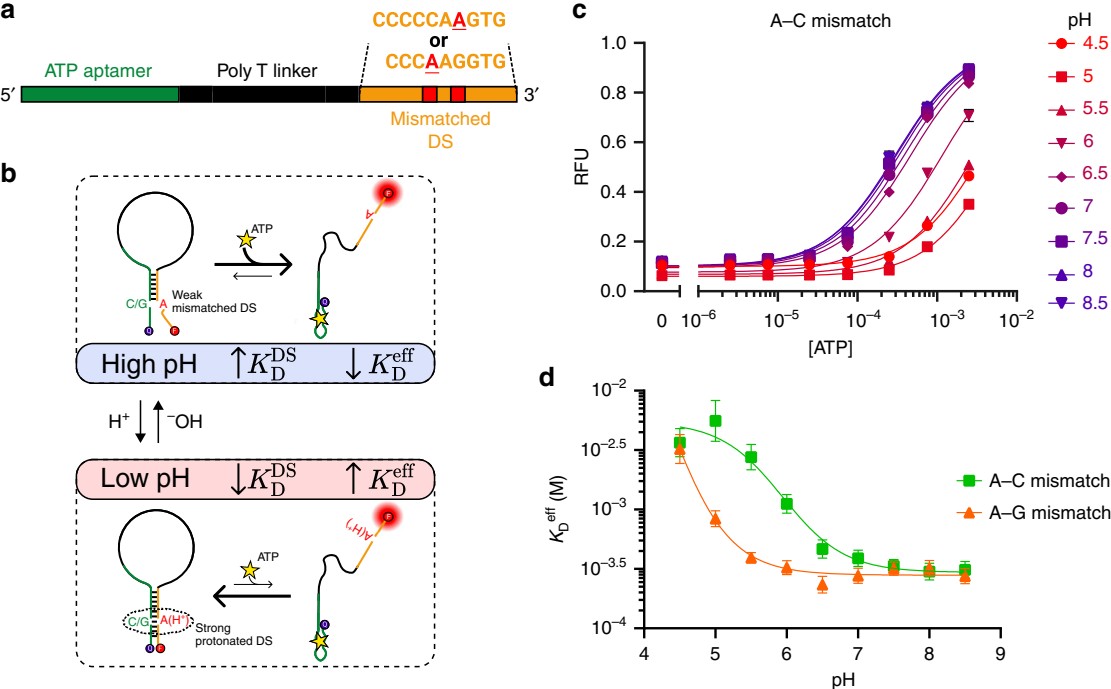

**Fig. 3 Rational design of a high pH-specific ATP aptamer. a** Preferential binding at neutral or alkaline pH is achieved by introducing an A–C or A–G mismatch into the DS sequence. **b** A simplified model for selective binding at higher pH. Mismatch protonation at acidic pH strengthens DS hybridization, decreasing the aptamer's effective affinity. A complete three-state model of the binding mechanism is given in Supplementary Fig. 5. **c** ATP binding curves for the A–C mismatch construct show increasing binding affinity at more alkaline pH. **d** Both A-C and A-G mismatches can be inserted to achieve stronger binding at neutral or alkaline pH than at acidic pH. Data points and error bars in **c** show the means and standard deviations of $n = 3$ independent experiments. Data points and error bars in **d** show the best fit values and standard deviations extracted from Langmuir isotherm fits to binding curve data from $n = 3$ independent experiments (data given in Supplementary Fig. 6) using nonlinear least squares fitting. Source data are provided as a Source Data file.

## Discussion

The ability to target medically important physiological environments using pH-responsive molecular switches could greatly improve specificity in diagnostic and therapeutic applications—for example, small variations in pH are key characteristics of the endosomal pathway and of cancerous tissue. However, the use of pH as a trigger for aptamer-based switches has been held back by the inherent challenge of engineering pH-responsive aptamers that can be tuned for optimal performance in a given physiological environment. Our PSD design allows us to rationally design pH-responsive molecular switches based on virtually any existing aptamer. Unlike previously described strategies, we impose minimal constraints on the pH-sensitive motifs within the PSD design, enabling rational fine tuning of both the magnitude of the pH response and the functional pH range of the switch. Our most sensitive constructs exhibit a pH response that greatly exceeds that of previously described aptamer switch designs, with up to a 1000-fold change in affinity between pH conditions. Our approach allows us to rapidly design constructs with minimal iteration and to predictably combine pH-responsive elements of individual switch designs in order to achieve more complex behavior, including a selective response to a narrowly defined window of pH conditions.

There are a number of promising potential applications for both low and high pH-specific binding that will benefit greatly from the ability to generate new target-specific molecular switches with tuned pH specificities. Healthy mammalian tissues typically have a physiological extracellular pH of ~7.4, whereas the tumor microenvironment is more acidic, with a pH of ~6.7–7.1[3]. Our low-pH-specific PSD design, which has a finely tunable $pK_a$ in the range of 6–7, would be highly valuable in the context of generating switches tailored for use in tumor imaging or therapeutic delivery. We also report what is, to the best of our knowledge, the first strategy for engineering affinity reagents that are selectively active within only a small window of pH conditions. Among other applications, such a construct could exert fine control over the cellular compartment in which an aptamer payload is released or retained. Biomolecules trafficked within the cell encounter steadily decreasing pH as they move from the extracellular matrix (pH ~7.4) to the early sorting endosome (pH ~6.2), with some ligands being recycled out of the cell and others moving to the late endosome (pH ~5.5) and then to the lysosome for degradation (pH < 5)[28]. Engineered reagents that are selectively active within narrow pH ranges could interact with ligands at specific stages of this trafficking process, probing or reprogramming endocytic function for therapeutic benefit[29].

Critically, because our strategy does not impose constraints that depend on the parent aptamer sequence, these complex functions could be implemented with virtually any aptamer. Strand displacement-based structure-switching designs have been successfully demonstrated for a range of parent aptamers, including aptamers with a wide range of affinities and those with complex secondary structures such as G-quadruplexes[7,30]. Our PSD design advances this approach while maintaining its broad applicability and the ability to rationally tune the resulting construct. We believe this high degree of generality will greatly expand the pool of possible biological targets and drug cargos for functional DNA nanodevices.

## Methods

**Reagents.** All oligonucleotide sequences (Supplementary Table 1) were synthesized by Integrated DNA Technologies with a 5′-end Iowa Black FQ quencher modification and a 3′-end Cy3 fluorophore modification. Oligonucleotides were resuspended at 100 μM in nuclease-free water before use. ATP (100 mM, 0.25 mL), UltraPure DNase/RNase-free distilled water, Tris-HCL (1 M, pH 8.0), sodium acetate (3 M, pH 5.5), UltraPure Bis-Tris, sodium hydroxide, and hydrochloric acid

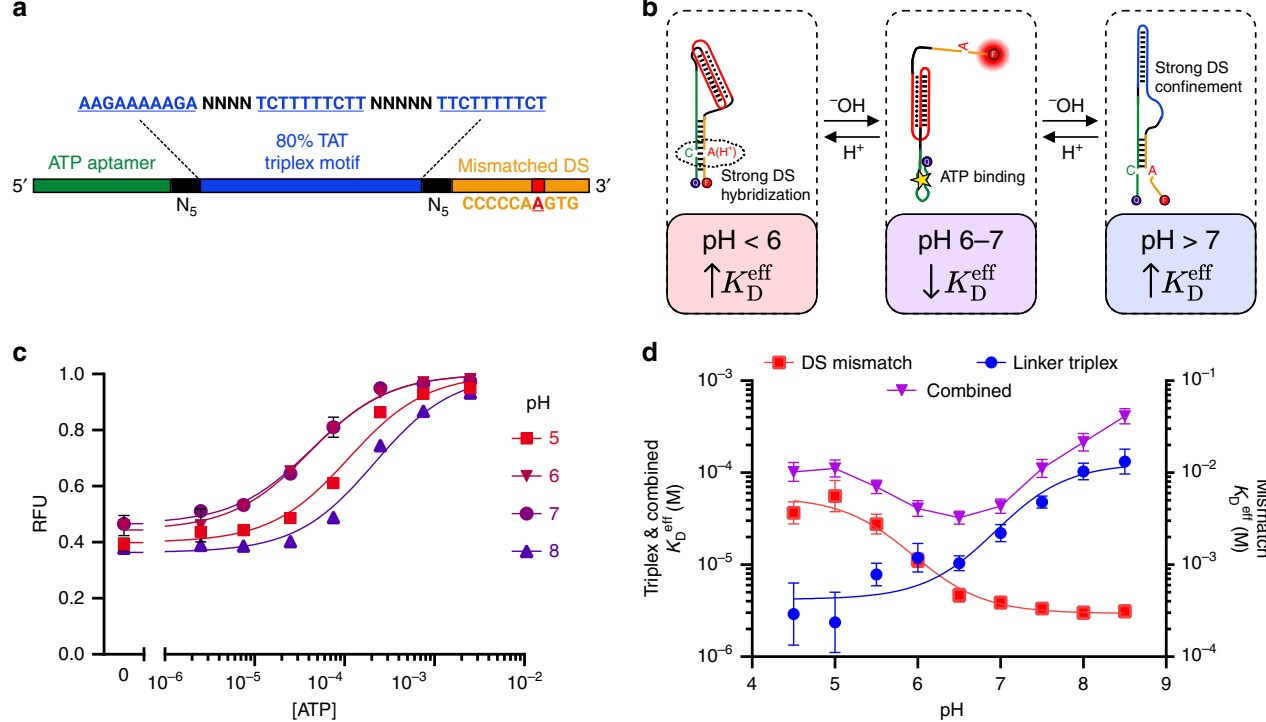

**Fig. 4 PSD design for selective binding over a narrow pH window. a** We designed a construct that incorporated both an intramolecular triplex in the linker domain and an A–C mismatch in the displacement strand. **b** Complex pH dependence is achieved through a three-state system, where binding is inhibited by strong DS hybridization at low pH or strong DS confinement at high pH, with a small window between these extremes where binding affinity is high. **c** ATP binding curves for the combined construct show the highest affinity at pH 6–7, with lower affinity at both high and low pH. **d** The pH dependence of affinity for this construct ($K_D^{eff}$ plotted on left axis) reflects contributions from both the binding inhibition of the linker modification-based TAT80 construct at high pH (left axis) and the binding inhibition of the DS modification-based A–C mismatch construct at low pH (right axis). Data points and error bars in **c** show the means and standard deviations of $n = 3$ independent experiments. Data points and error bars in **d** show the best fit values and standard deviations extracted from Langmuir isotherm fits to binding curve data from $n = 3$ independent experiments (data given in Supplementary Fig. 7) using nonlinear least squares fitting. Source data are provided as a Source Data file.

were purchased from Thermo Fisher Scientific. Sodium azide was purchased from Sigma Aldrich.

**Sequence optimization.** To minimize unintended secondary structure created by inserting the triplex motif into the PSD construct linker domain, we structured our construct to contain two five-nt spacers, and optimized these spacer sequences along with the triplex loop sequences using the NUPACK tubedesign algorithm[20]. Construct sequences were constrained to be of the form:

5′-(ATP aptamer)-$N_5$-(Triplex)-$N_4$-(Triplex)$^*$-$N_5$-(Triplex)′-$N_5$-(DS)-3′.

Where the constituent sequences are:

(ATP Aptamer): 5′-CACCTGGGGGGAGTATTGCGGAGGAAGG-3′.

(DS): 5′-CCAGGTG-3′ for triplex constructs, and 5′-CCCCCAAGTG-3′ for bandpass constructs.

(Triplex): Various 5′-RRRRRRRRRR-3′ sequences, where R = A or G, with selected bases forced to A or G based on desired % TAT content.

(Triplex)$^*$ and (Triplex)′ are, respectively, the reverse-complement and complement of the (Triplex) sequence.

The secondary structure optimization target was formation of a seven-nt hairpin between the 5′ end of the ATP aptamer and the DS, and a ten-nt hairpin between the (Triplex) and (Triplex)$^*$ sequences, with all other regions forming unpaired loops. Target secondary structure and resulting secondary structures after optimization are shown in Supplementary Fig. 8. The $N_4$ and $N_5$ regions were then optimized under solution conditions of 1 μM construct, 50 mM Na$^+$, and 6 mM Mg$^{2+}$ at 25 °C, with a stop condition of 5% ensemble secondary structure defect. Not all constructs reached the desired stop condition. In these cases, multiple design outputs were compared and the design with lowest minimized ensemble defect was selected. The final constructs obtained through optimization, as well as the sequence constraints used in optimization, are given in Supplementary Table 1.

**Binding affinity measurements.** To ensure consistent sodium ion content between all pH values, we formulated a universal binding buffer (10 mM Tris-HCL, 10 mM Bis-Tris, 10 mM sodium acetate, 6 mM MgCl$_2$, 0.01% sodium azide) adjusted to pH 8.55 with sodium hydroxide. We then adjusted individual aliquots of buffer to the desired final pH (4.5–8.5, increments of 0.5) with hydrochloric acid.

To obtain binding curves, we prepared 50 μL samples of 250 nM aptamer in buffer at each pH, then added 5 μL of ATP in nuclease-free water to bring the final ATP concentration into the range of 0–7.5 mM. Mixtures of aptamer and target were incubated at room temperature for 30 min. Fluorescence was measured on a Synergy H1 microplate reader (BioTeK) with Cy3 excitation (538-nm center wavelength, 63-nm bandpass width) and emission (590-nm center wavelength, 35-nm bandpass width) filters and extended dynamic range measurement. All measurements were taken in triplicate.

**Analysis of pH-dependent binding properties.** All data analysis was performed in GraphPad Prism 8.0.2. Raw triplicate Cy3 fluorescence ($F_{Cy3}$) vs. ATP concentration ([ATP]) data for each construct and pH combination were fit to a Langmuir isotherm binding model of the form:

$$F_{Cy3} = (A - y_0) \frac{[\text{ATP}]}{10^{pK_D} + [\text{ATP}]} + y_0, \tag{1}$$

where $pK_D = \log_{10}\left(K_D^{eff}\right)$ is used to extract the binding affinity of the construct at the given pH, $y_0$ is the background signal from the construct at [ATP] = 0, and $A$ is the maximum signal from the construct at saturating target concentration. We corrected for small variations in aptamer concentration between different pH binding curves for the same construct, as well as pH dependencies in fluorophore-quencher system intensity between pH conditions, by normalizing all binding curves to their maximum saturated signal ($A$). Normalized binding curves were produced by correcting all data to $F_{Norm} = \frac{F_{Cy3}}{A}$, followed by replotting with a Langmuir isotherm fit that was normalized by the same factor $A$. Binding affinities are reported as the best fit value with 95% confidence interval upper and lower bounds from fitting.

The pH-dependent properties of each construct were quantified by evaluating changes in the dissociation constant $pK_D = \log_{10}\left(K_D^{eff}\right)$ as a function of pH. When fitting to this data, we used the standard error of the $pK_D$ extracted from the Langmuir isotherm fit to each construct/pH condition as the standard error for each corresponding data point. We fit pH-dependence curves with the functional form:

$$pK_D = \Delta pK_D \frac{10^{-pH}}{10^{-pK_a} + 10^{-pH}} + pK_{D,High}, \tag{2}$$

where $pK_a$ is the transition midpoint of the construct's pH response, and is reported as the best fit values with 95% confidence interval upper and lower bounds from fitting. Then we extracted the properties of our construct in terms of dissociation constants based on the following relationships:

$$pK_{D,High} = \log_{10}\left(K_{D,High}^{eff}\right), \tag{3}$$

and

$$\Delta pK_D = \log_{10}\left(\Delta K_D^{eff}\right) = \log_{10}\left(K_{D,Low}^{eff} - K_{D,High}^{eff}\right). \tag{4}$$

Based on these calculated values, $\Delta K_D^{eff}$ serves as a measure of the fold change in pH response over the functional range.

**Reporting summary**. Further information on research design is available in the Nature Research Reporting Summary linked to this article.

## Data availability

All data underlying the findings of this study are available from the authors upon reasonable request. The source data underlying Figs. 1–4 and Supplementary Figs. 2, 4, 6, and 7 are provided as a Source Data file. Source Data are provided with this paper.

## Code availability

The Nupack code used to optimize triplex-containing sequences is provided at: https://github.com/ianapt/PSD_design.

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

## Acknowledgements
This work was supported by the Chan-Zuckerberg Biohub. I.A.P.T. was supported by the Medtronic Foundation Stanford Graduate Fellowship. We thank Oğuz Tolga Çelik for his assistance in performing binding affinity measurements. We also thank Dr. Evelin Sullivan of the Technical Communications Program at Stanford for her thoughtful comments and edits on the paper.

## Author contributions

I.A.P.T. and H.T.S. devised the initial concept. I.A.P.T. and L.Z. designed aptamer constructs. I.A.P.T designed experiments, executed experiments, and analyzed the data. I.A.P.T., M.E., and H.T.S. wrote the paper. All authors edited and discussed the paper.

## Competing interests

The authors declare no competing interests.
