## [Peer Review File · Nature Communications]

Reviewers' Comments:

Reviewer #1:

Remarks to the Author:

The manuscript "Rational Design of Aptamer Switches with Programmable pH Response" presents an interesting approach to achieve pH regulation of DNA-based aptamers. The authors took inspiration from previous works on triplex based DNA systems (ref. 11 and 19) in which triplex DNA have been used for creating pH-dependent nanoswitches with tunable pKa and present a novel approach to tune aptamer's affinity with pH focusing on the well known ATP binding aptamer. The manuscript is very clear and the results are convincing and conclusions justified so in my opinion the manuscript deserves publication provided that minor issues below are addressed.

1) The rationale of the general approach is very clever and interesting. It is somewhat related to what was presented in ref. 11 by Ricci group. The authors rightly cited this and other previous literature records. However, the difference between the approach proposed in ref 11 and this article should be better described.

The manuscript regarding pH controlled DNA strand displacement (JACS, 2014, 136, 16469) seems also relevant for the strategy reported here and should be cited.

2) Figure 1: use of colors in this figure (panel a) was quite confusing for me. Authors should revise and improve clarity. Same applies to Figure 4. The change of color from duplex to triplex is confusing for the reader, I think. I would use another way to show the folding/unfolding.

3) Use of too many significant figures in the definition of KD and pKa should be avoided. Please use only one significant figure after the comma.

4) It is not clear what the authors mean with this sentence. "more granular tuning of pH response in this system relative to the range achieved by Ricci et al. in the tuning of standalone triplex". Authors should improve clarity of this sentence.

5) Authors are studying the effect of the ATP affinity on the pH variation. This is not completely the same as done in ref. 19 where pH titration curves (independent on ligand binding) were described. It would be interesting to show similar pH titration curve to actually see the folding/unfolding of the triplex at different pHs. This could be a better comparison with ref. 19 results. The "more granular tuning" described in the article could still be present but in this case would be more readily comparable. Authors could add this in their results section (or maybe even supplementary info).

6) "replacing TAT triplets with more stable CGC triplets increases the strength of triplex formation, thereby increasing the magnitude of the pH response [20,21]".

This sentence is not clear to me. Why this should be the case? Why TAT triplets should be more stable? What the authors mean here is probably that TAT are less pH-dependent. Please improve clarity of this sentence.

7) Figure 3, panel d: please revise y-axis label.

Francesco Ricci

Reviewer #2:

Remarks to the Author:

The work describes an interesting and potentially general approach to convert an aptamer into a pH-responsive switch that can be tuned to selectively undergo a dramatic shift in affinity under acidic, neutral, or alkaline conditions. The approach uses two orthogonal modes of control that can

be manipulated in parallel: 1) incorporation of an intramolecular triplex motif and 2) strategic insertion of pH sensitive A-G or A-C mispairs.

This approach is built on existing knowledge but is very novel in the combination of the two orthogonal strategies. I think there will be a lot of interest in this approach. The work performed is meticulous and convincing. My only concerns are ones that are difficult in our current lockdown situation so I'm suggesting only some additional discussion.

1) How can we be sure this is general when it is tested on just one aptamer system? Perhaps, rather than ask for new experiments, the authors can address this in a more fulsome manner in the discussion. How would this be adapted to say an g-quad based aptamer, or one that already contains some triplex motif?

2) The ATP aptamer is already not a very tightly binding aptamer - perhaps making it have lower affinity for a target is not that difficult. Some discussion around how this would be applicable to tighter binders would be helpful.

RE: NCOMMS-20-10207

Author Responses to Reviewers:

Reviewer #1: The reviewer was positive about our work, noting the novelty of our approach and clarity of our results, and suggesting publication upon addressing some minor comments. The review comments raised important points on how the clarity of our approach could be improved and how we could better frame our results within the existing body of literature. We thank the reviewer for these valuable comments, which are addressed below:

- 1. The reviewer notes: “The rationale of the general approach is very clever and interesting. It is somewhat related to what was presented in ref. 11 by Ricci group. The authors rightly cited this and other previous literature records. However, the difference between the approach proposed in ref 11 and this article should be better described. The manuscript regarding pH controlled DNA strand displacement (JACS, 2014, 136, 16469) seems also relevant for the strategy reported here and should be cited.”**

We agree with the reviewers observation that we could better highlight the differences between our approach and the work previously demonstrated by the Ricci group¹. We have updated the introduction to reflect the key elements that differ between our approach and prior pH-responsive aptamer designs. Notably, the introduction of our selected pH motifs into the non-binding linker or DS domains, rather than in direct competition with the native aptamer structure, minimizes sequence constraints imposed by the aptamer to allow broader tunability of the design.

Additionally, we appreciate the reviewer’s suggestion and have included this reference in the updated manuscript.

- 2. The reviewer advises: “Figure 1: use of colors in this figure (panel a) was quite confusing for me. Authors should revise and improve clarity. Same applies to Figure 4. The change of color from duplex to triplex is confusing for the reader, I think. I would use another way to show the folding/unfolding.”**

We thank the reviewer for this suggestion and agree that because the constructs and schemes are complex, their clarity is critical. However, we feel that using color of the construct to depict the duplex-to-triplex transition is a useful shorthand and visual indicator of pH condition. To make sure this shorthand is clear, we have added additional detail to Figure 1 (panel b) to denote the meaning of the color scheme in triplex folding. We have also modified the colors used to label the triplex motif in panel a to more clearly connect the described construct to those shown in panel b. We hope that with this early clarification, Figure 4 will also be clearer for the reader.

- 3. The reviewer requests: “Use of too many significant figures in the definition of KD and pKa should be avoided. Please use only one significant figure after the comma.”**

We agree with the reviewer and apologize for the oversight in previous reporting of significant figures. We have updated the manuscript throughout to correct our reported values to have two significant figures in all cases.

4. The reviewer suggests: “It is not clear what the authors mean with this sentence. ‘more granular tuning of pH response in this system relative to the range achieved by Ricci et al. in the tuning of standalone triplex.’ Authors should improve clarity of this sentence.”

We agree with the reviewer’s suggestion that this statement could be made clearer. To clarify, our observation is that the shift in pK_a induced by a single TAT/CGC substitution within the triplex motif appears to be smaller for our constructs than previously reported values from standalone intramolecular triplex motifs². We believe that this smaller per-substitution shift in pK_a should offer finer-grained tuning of the construct’s pK_a as it can be adjusted in smaller increments. We have revised our explanation in the manuscript to more clearly reflect this observation.

5. The reviewer requests: “Authors are studying the effect of the ATP affinity on the pH variation. This is not completely the same as done in ref. 19 where pH titration curves (independent on ligand binding) were described. It would be interesting to show similar pH titration curve to actually see the folding/unfolding of the triplex at different pHs. This could be a better comparison with ref. 19 results. The “more granular tuning” described in the article could still be present but in this case would be more readily comparable. Authors could add this in their results section (or maybe even supplementary info).”

We agree with the reviewers that pH titration curve data in the absence of ligand could provide an additional measure of the pH-dependence of triplex folding. This in turn could better support our assertion that the tuning of pK_a is more granular within our system. Fortunately, this data can be extracted from our existing dataset by considering only the ligand-free ($[ATP] = 0$ M) datapoints to obtain pH titration curves for each triplex construct. Normalizing and analyzing this pH titration data to extract the pK_a for ligand-free switching, we recover the general trend that increased triplex TAT content increased the switch pK_a and confirm that the range of pK_a shift when moving from 50% to 80% TAT content is much narrower than observed standalone triplexes in prior work² (results shown in figures below).

We have updated the manuscript to include this data in **Supplementary Figure 2** and have updated our discussion of “granular tuning” in the main text to present this more direct comparison of pH titration results.

6. The reviewer comments: “replacing TAT triplets with more stable CGC triplets increases the strength of triplex formation, thereby increasing the magnitude of the pH response [20,21].’ This sentence is not clear to me. Why this should be the case? Why TAT triplets should be more stable? What the authors mean here is probably that TAT are less pH-dependent. Please improve clarity of this sentence.”

The reviewer is correct in their interpretation. Our intention was to communicate that the stability of TAT triplet formation is less pH-dependent as compared to CGC triplets, leading to increased pH-response magnitude with increased CGC content. This explanation has been revised in the text.

7. The reviewer suggests: “Figure 3, panel d: please revise y-axis label.”

We thank the reviewer for catching this error. The axis labeling has been revised in the manuscript.

Reviewer #2: The reviewer was enthusiastic about the novelty, potential impact, and quality of our presented results, stating that “this approach is built on existing knowledge but is very novel in the combination of the two orthogonal strategies. I think there will be a lot of interest in this approach. The work performed is meticulous and convincing.” We thank the reviewer for their insightful suggestions on how additional discussion could strengthen the key conclusions of our work. We have addressed these suggestions as follows:

1. The reviewer asks: “How can we be sure this is general when it is tested on just one aptamer system? Perhaps, rather than ask for new experiments, the authors can address this in a more fulsome manner in the discussion. How would this be adapted to say an g-quad based aptamer, or one that already contains some triplex motif?”

We appreciate the reviewers concern, as the utility of this design is dependent upon its generality. A key point in our strategy was basing our design on the ISD aptamer design, which has been demonstrated to work with a range of aptamers from less structured aptamers (ATP), to G-quadruplex containing aptamers (Thrombin)³. The design of ISD structure-switching aptamers for a broad variety of ligands is discussed by Munzar et al. as part of a comprehensive review (ISD aptamers are referred to as cis-DA aptamers in this work; Table 1 is a key resource that lays out the various existing aptamers)⁴. In brief, successful conversion of a parent aptamer to an ISD switch relies on selecting a region of the aptamer to which the displacement strand can hybridize and inhibit target binding, a functionality which can be achieved through screening. We believe this can be implemented successfully with a broad range of aptamers, and thus the same should be true for our PSD design.

These important points have been added to the discussion section of the manuscript.

2. The reviewer suggested: “The ATP aptamer is already not a very tightly binding aptamer - perhaps making it have lower affinity for a target is not that difficult. Some discussion around how this would be applicable to tighter binders would be helpful.”

We thank the reviewer for their question. Because our design uses an intramolecular displacement strand (DS) to compete with ligand binding, its behavior is that of a three-state population shift equilibrium (Supplementary Figure 1). Notably, for this equilibrium, the observed target binding affinity is:

$$K_D^{eff} = K_D^{apt} (1 + K_Q)$$

Where K_D^{apt} is the affinity of the native aptamer, while K_Q is an equilibrium constant reflecting the strength of DS competition⁵. Our construct achieves pH-responsiveness through pH-dependent shifts in K_Q that shift the affinity of the construct *relative* to the aptamer’s native affinity. Thus, application of this strategy with a higher affinity aptamer (lower K_D^{apt}), will produce designs that bind with higher affinity, but the *relative* shifts in binding affinity will be the same, since they are driven by K_Q and independent of

the choice of parent aptamer. This means that even for higher affinity aptamers, we should expect the same degree of selectivity in binding (in terms of fold-change in affinity) between high and low pH when using the PSD design.

We feel that while this discussion is useful, much of the underlying technical detail is presented in our lab's previous investigation of these constructs⁵. Thus, we have included a brief statement on the generality of our approach for aptamers with a wide range of affinities, but have omitted the complete technical explanation given above.

References

1. Porchetta, A., Idili, A., Vallée-Bélisle, A. & Ricci, F. General Strategy to Introduce pH-Induced Allosterity in DNA-Based Receptors to Achieve Controlled Release of Ligands. *Nano Lett.* **15**, 4467–4471 (2015).
2. Idili, A., Vallée-Bélisle, A. & Ricci, F. Programmable pH-Triggered DNA Nanoswitches. *J. Am. Chem. Soc.* **136**, 5836–5839 (2014).
3. Tang, Z. *et al.* Aptamer Switch Probe Based on Intramolecular Displacement. *J. Am. Chem. Soc.* **130**, 11268–11269 (2008).
4. Munzar, J. D., Ng, A. & Juncker, D. Duplexed aptamers: history, design, theory, and application to biosensing †. *Chem. Soc. Rev* **48**, 1390 (2019).
5. Wilson, B. D., Hariri, A. A., Thompson, I. A. P., Eisenstein, M. & Soh, H. T. Independent control of the thermodynamic and kinetic properties of aptamer switches. *Nat. Commun.* **10**, 5079 (2019).